# INGENIOUS: Using Informative Data Subsets for Efficient Pre-Training of Language Models

**H S V N S Kowndinya Renduchintala**[1][*] **Krishnateja Killamsetty**[2]**, Sumit Bhatia**[3]
**Milan Aggarwal**[3]**, Ganesh Ramakrishnan**[1]**, Rishabh Iyer**[2]**, Balaji Krishnamurthy**[3]
[1] Indian Institute of Technology Bombay, India
[2] University of Texas at Dallas, USA
[3] Media and Data Science Research (MDSR) Lab, Adobe Inc., India

## Abstract

A salient characteristic of pre-trained language models (PTLMs) is a remarkable improvement in their generalization capability and emergence of new capabilities with increasing model capacity and pre-training dataset size. Consequently, we are witnessing the development of enormous models pushing the state-of-the-art. It is, however, imperative to realize that this inevitably leads to prohibitively long training times, extortionate computing costs, and a detrimental environmental impact. Significant efforts are underway to make PTLM training more efficient through innovations in model architectures, training pipelines, and loss function design, with scant attention being paid to optimizing the utility of training data. The key question that we ask is whether it is possible to train PTLMs by employing only highly informative subsets of the training data while maintaining downstream performance? Building upon the recent progress in informative data subset selection, we show how we can employ submodular optimization to select highly representative subsets of the training corpora and demonstrate that the proposed framework can be applied to efficiently train multiple PTLMs (BERT, BioBERT, GPT-2) using only a fraction of data. Further, we perform a rigorous empirical evaluation to show that the resulting models achieve up to $\sim 99\%$ of the performance of the fully-trained models. We made our framework publicly available at https://github.com/Efficient-AI/ingenious.

## 1 Introduction

Pre-trained language models (PTLMs) (Devlin et al., 2019; Radford et al., 2019; Yang et al., 2020; Brown et al., 2020; Raffel et al., 2020) have revolutionized the field of natural language processing (NLP), becoming the default choice for a wide array of NLP tasks. The versatility of PTLMs,

---

[*]Work done during internship at Media and Data Science Research (MDSR) Lab, Adobe Inc.

however, is accompanied by significant costs. For instance, it costs an estimated $12 million to train GPT-3 (Brown et al., 2020) with roughly 1.2 million pounds of $CO_2$ emissions (Kahn, 2021). Megatron-Turing NLG (Smith et al., 2022) is a 530 billion parameter PTLM, which is thrice the size of GPT-3 and is trained on 4480 A100 GPUs and yields close to 1% performance improvements over GPT-3. By continually increasing the size of PTLMs and pre-training corpora to improve generalization ability, significant additional resources and energy are consumed, resulting in dire environmental consequences (Sharir et al., 2020). Further, such large-scale resource utilization and the costs associated with PTLMs create an uneven playing field for small organizations and universities, which operate with significant resource constraints. Hence, a crucial step towards developing responsible, fair, and GreenAI (Schwartz et al., 2020) involves minimizing inefficiencies and costs of training these models.

Significant efforts toward improving the efficiency of PTLMs have ventured in directions such as optimizing the model architecture (Chen et al., 2020; Gordon et al., 2020; Zafrir et al., 2021), modifications to the training pipeline (Izsak et al., 2021; Shen et al., 2022) and task (Schick and Schütze, 2021), sample efficient masking techniques for improved convergence (Bitton et al., 2021) and leveraging contextual knowledge to reduce model size (Kaur et al., 2022). In this work, driven by the observation that the scale of the pre-training corpus contributes significantly to the training costs of PTLMs, we explore the feasibility of training PTLMs using highly informative subsets of the corpus. Recent studies have demonstrated the feasibility of informative data subset selection for efficient deep model training for images (Mirzasoleiman et al., 2020; Killamsetty et al., 2021a,b,c; Pooladzandi et al., 2022) in both supervised and

semi-supervised settings. In light of this, the key question we attempt to answer is: *Can we efficiently pre-train language models using highly informative subsets of the training corpus without compromising performance?*

The first step in answering the above question is identifying informative (or representative) subsets of the underlying training corpus such that they maximize the representation of the remaining samples in the corpus. Intuitively, given a set of sentences, the subsequent addition of sentences similar to existing sentences in the set yields *diminishing returns*. More information gains can be achieved by adding diverse, dissimilar sentences. While the classical subset selection problem is NP-hard, we can leverage the *diminishing gains* property of submodular functions (Fujishige, 2005) and frame subset selection as a submodular maximization problem. Several recent works (Wei et al., 2015; Mirzasoleiman et al., 2020; Kothawade et al., 2021; Karanam et al., 2022; Maheshwari et al., 2020) have formulated the subset selection problem as that of maximizing a submodular objective. However, applying existing subset selection frameworks to PTLMs is nontrivial given the scale of corpora typically used for pre-training (*e.g.*, Wikipedia and Common Crawl consisting of hundreds of millions of sequences and billions of tokens). Most of the existing methods rely on per-sample gradients, which are expensive to compute, and to the best of our knowledge, none of the previous works have considered subset selection for such large datasets.

**Our contributions:** We propose the informative data subset selection task for efficient pre-training of PTLMs and present INGENIOUS, a framework for subset selection using submodular optimization (Section 3). We show how to overcome the scalability challenge for typical large-scale pre-training corpora and employ scalable sentence feature encoders to obtain individual data sample features relevant for subset selection. We also employ various engineering techniques to scalably select subsets from large-scale datasets (Section 3). We use INGENIOUS to pre-train BERT and GPT-2 and evaluate the performance of the resulting models on downstream tasks (Section 4). A rigorous empirical evaluation reveals that the models pre-trained with INGENIOUS retain upto ≈ 99% performance of the models pre-trained using the

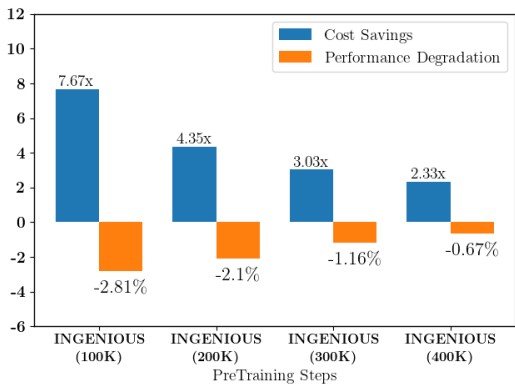

Figure 1: Cost-savings vs Performance tradeoff achieved by INGENIOUS for BERT pre-training: We contrast the accuracy degradation with cost savings compared to the vanilla BERT pre-training on entire dataset. We observe $4.35\times$ cost-savings with $2.1\%$ accuracy drop and $2.33\times$ cost-savings with $0.67\%$ accuracy drop.

full dataset. Figure 1 summarizes the cost-savings vs performance trade-off achieved by INGENIOUS for BERT pre-training. We also present thorough ablation studies revealing the impact of various design choices and parameters involved. We also evaluate the models trained by INGENIOUS in terms of their knowledge retention capabilities and show how INGENIOUS can be used to accelerate pre-training of domain-specific language models such as BioBERT (Section 4.4). Finally, we discuss the inferences that could be drawn from our work, limitations of our proposed framework and lay out directions for further improvement (Section 5).

## 2 Related Work

***Knowledge distillation and pruning based methods*** (Sanh et al., 2019; Jiao et al., 2020; Muhamed et al., 2021) pre-train a smaller variant of PTLMs (such as BERT) with lesser capacity using the full model as teacher network. Even though lighter versions such as DistilBERT (Sanh et al., 2019) retain ≈ 97% of the performance with up to 60% faster inference, the PTLM still needs to be ***completely*** pre-trained initially to be able to distill the lighter version. Thus, the efficiency gains are restricted only to the fine-tuning and inference. Other methods prune the architecture through forcing the weights with lesser magnitude to zero value during pre-training (Chen et al., 2020; Gordon et al., 2020) as well as during fine-tuning (Zafrir et al., 2021).

*Model architecture and training task optimizations:* Schick and Schütze (2021) have shown that smaller PTLMs can achieve better performance by formulating the task input in cloze style. Izsak et al. (2021) proposed to optimize BERT pre-training through multiple optimizations related to data, model size, and optimizer choice. Shen et al. (2022) proposed a staged training mechanism where they start with training a relatively smaller model, which is then used for initializing the full capacity model at a later stage. Yao et al. (2022) identify relevant samples from the pre-training corpus based on their similarity with the task-specific dataset to train task-specific PTLM followed by fine-tuning, thus inherently suffering from the limitation of pre-training separate models for every downstream task.

*Curriculum learning based methods* employ the sequence length of training samples as a proxy for hardness.Typically, shorter (easier) sequences are presented in the initial stages of pre-training followed by longer (harder) sequences at later stages (Nagatsuka et al., 2021; Li et al., 2022). However, such methods have been shown to perform well only in limited configurations with respect to the choice of language models, stage of pre-training, *etc.*.

**Hardware optimizations for PTLM Training:** The suite of Open Pre-Trained Transformers (OPT) (Zhang et al., 2022) require 1/7th of the carbon footprint for pre-training when compared to popular PTLMs such as GPT-3 (Brown et al., 2020) while achieving comparable few-shot generalization. OPTs leverage extensive data and tensor parallelism with high-memory GPUs (supporting large batch sizes), which are usually not easily accessible and can lead to exorbitant costs.

Noticeably different from the aforementioned works, we explore making PTLM training more efficient by utilizing highly informative subsets of the training data. Consequently, our proposal effectively complements other optimization methods that target aspects such as model architecture and hardware enhancements.

## 3   The INGENIOUS Framework

We now present INGENIOUS - an informative data subset selection framework for pre-training language models. We summarize the training pipeline in Figure 2. We first describe the nota-

tion to formulate the problem, followed by details of different steps involved in the framework.

### 3.1   Notation

We denote the unlabeled dataset for pre-training by $\mathcal{U} = \{x_j\}_{j=1}^n$, consisting of $n$ data points each corresponding to a varying length of sequence of symbols $\{s_i\}_{i=1}^m$ (these symbols could be words or character sequences such as sub-words). Let $\mathcal{S} \subseteq \mathcal{U}$ be the subset of the unlabeled dataset on which the language model is trained. Let the language model be parameterized by $\boldsymbol{\theta}$. We subscript the changing variables such as model parameters $\boldsymbol{\theta}$, subset $\mathcal{S}$ with the timestep $t$ to denote their specific values at that timestep.

### 3.2   Problem Formulation

In its most general form, subset selection is defined as
$$\mathcal{S}_t = \arg\max_{\mathcal{S} \subseteq \mathcal{U}} f(\mathcal{S}) \qquad (1)$$
where the subset $\mathcal{S}_t \subseteq \mathcal{U}$ at step $t$ is selected such that it maximizes the function $f$.

While the above general subset selection problem is NP-Hard, the problem becomes approximable in case the function $f$ is submodular in nature (Fujishige, 2005). A set function $f : 2^{\mathcal{U}} \to \mathbb{R}$ is **submodular** if for $x \in \mathcal{U}$, $f(\mathcal{A} \cup x) - f(\mathcal{A}) \geq f(\mathcal{B} \cup x) - f(\mathcal{B})$, $\forall \mathcal{A} \subseteq \mathcal{B} \subseteq \mathcal{U}$ and $x \notin \mathcal{B}$. We pose the data subset selection problem as a submodular maximization problem since it allows for easier optimization by employing different approximations (Nemhauser et al., 1978; Iyer and Bilmes, 2019). In order to choose a suitable submodular function, one must understand the characteristics of the subsets that are crucial for the end-goal – *efficient learning in our case*. Previous works in computer vision have demonstrated that commonly used vision datasets contain many redundancies, and eliminating such redundant data samples does not affect the model's performance (Birodkar et al., 2019; Toneva et al., 2019; Paul et al., 2021; Sorscher et al., 2022). Further, one can achieve faster model training by using highly informative and representative data subsets (Kaushal et al., 2019; Mirzasoleiman et al., 2020; Sorscher et al., 2022). Please refer to Appendix B for more related work on submodularity based subset selection. Building upon the learnings from computer vision research, our primary requirement for the selected subset is that it should faithfully represent the training data and have min-

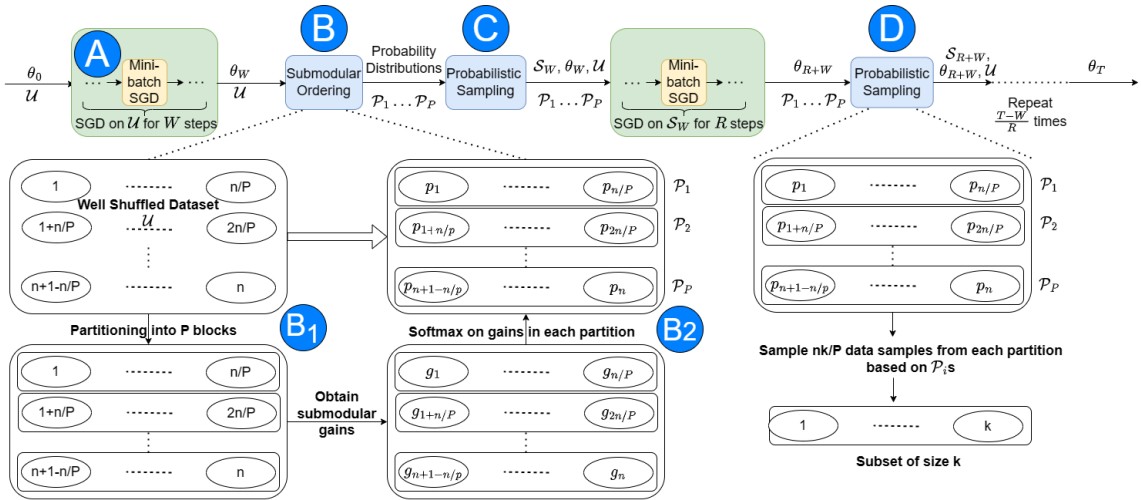

Figure 2: INGENIOUS framework for informative data subset selection to pre-train language models. We warm-start pre-training for $W$ steps to enable it to learn useful representations (step $A$). Owing to the size of pre-training data, we divide the total number of samples (n) into P partitions (step $B_1$) followed by selecting instances according to submodular gains (step $B_2$) through probabilistic sampling (step $C$). We obtain a subset (of total size k) of representative samples from each partition such that the subset is updated periodically (step $D$) after R steps of training on selected subset.

imal redundancy within itself.

## 3.3 Overview of Approach

In order to select a representative subset as discussed above, we use **Facility Location** (Salhi, 1991; Krause and Golovin, 2014), a commonly-used submodular function closely related to $k$-medoid clustering which is defined as

$$f_{FL}(\mathcal{A}) = \sum_{i \in \mathcal{U}} \max_{j \in \mathcal{A}} \mathcal{K}_{ij} \qquad (2)$$

where $\mathcal{A}$ is the subset being evaluated, $\mathcal{K}$ is a pairwise similarity matrix and $\mathcal{K}_{ij}$ is the similarity between the $i^{th}$ and $j^{th}$ samples. Thus, our subset selection problem can be represented as:

$$\mathcal{S}_t = \underset{\mathcal{S} \subseteq \mathcal{U}:|\mathcal{S}|=k}{\arg\max} f_{FL}(\mathcal{S}) \qquad (3)$$

Here, $k$ represents the size of the subset $\mathcal{S}$. We would like to clarify that Equation (3) enables us to choose diverse samples such that each represents other samples in the corpus, instead of selection of similar samples. The optimization problem in Equation (3) is an instance of cardinality-constrained monotone submodular maximization for which an approximate solution can be obtained by incrementally building the subset from scratch using algorithms such as Naive Greedy (Nemhauser et al., 1978), Lazy Greedy (Minoux, 1978), Stochastic Greedy (Mirzasoleiman et al., 2015), Lazier-than

Lazy-Greedy (Mirzasoleiman et al., 2015). We use the Lazier-than-Lazy Greedy optimizer as it is the most computationally efficient, along with memoization (Iyer and Bilmes, 2019).

The facility location function utilizes a pairwise similarity kernel $\mathcal{K}$ (of size $|\mathcal{U}| \times |\mathcal{U}|$) between the data samples in $\mathcal{U}$ to select representative subsets. To estimate the kernel values, we compute the cosine similarity between the feature representations of data samples obtained using the LM itself. To ensure that the extracted representations are meaningful during the initial phase, we warm start the model for W training steps as suggested by Killamsetty et al. (2021a,c) (step $A$ in Figure 2) . Further, to ensure that LM sees diverse data samples, we probabilistically sample data points based on submodular ordering obtained from running the greedy algorithm(steps $B$ and $C$ in Figure 2) and update the subset after every $R^{th}$ iteration (step $D$ in Figure 2) .

This re-sampling procedure is repeated till the pre-determined number of steps. Algorithm 1 summarises the steps involved and in the following section, we describe the details of each step.

## 3.4 Methodology Details

**Feature Encoders for Similarity Computation:** The selection of optimal representative subsets requires a similarity kernel that captures the intrinsic relationships between data samples. We ex-

**Algorithm 1:** Pre-Training using INGE-NIOUS

**Input:** Training dataset: $\mathcal{U}$, Initial model parameters: $\theta_0$, Total no of training steps: $T$, Training steps interval for subset selection: $R$, Number of steps for warmstart phase: $W$, Size of the subset: $k$, Learning rates: $\{\alpha_t\}_{t=0}^{t=T-1}$

Set $t = 0$
optimizer = AdamW()
*** Warmstart Phase ***
**repeat**
    Compute batches $\mathcal{U}_b = ((x_b, y_b); b \in (1 \cdots B))$ from $\mathcal{U}$
    **for** $b = 1$ *to* $B$ **do**
        **if** $t \geq W$ **then**
            break
        Compute mask $\mathbf{m}_t$ on $\mathcal{U}_b$
        $\theta_{t+1}$ = optimizer.step()
        $t = t + 1$
**until** *until* $t \geq W$
*** Subset Selection ***
greedyIdxs, gains $= argmax_{|S| \leq |\mathcal{U}|} f_{FL}(S, \mathcal{U}, \theta_t)$
probabilities = TaylorSoftmax(gains)
$\mathcal{S}_t \sim$sample(greedyIdxs, probabilities, k)
**repeat**
    Compute batches $\mathcal{S}_{tb} = ((x_b, y_b); b \in (1 \cdots B))$ from $\mathcal{S}_t$
    **for** $b = 1$ *to* $B$ **do**
        **if** $t \geq T$ **then**
            break
        Compute mask $\mathbf{m}_t$ on $\mathcal{S}_{tb}$
        $\theta_{t+1} = optimizer.step()$
        $t = t + 1$
        **if** $(t \% R == 0)$ **then**
            $\mathcal{S}_{t+1} \sim$sample(greedyIdxs, probabilities, k)
            break
        **else**
            $\mathcal{S}_{t+1} = \mathcal{S}_t$
**until** *until* $t \geq T$
*** Evaluate trained model on validation set ***
$eval$ = evaluate $(\theta_T, \mathcal{V})$
**return** $eval$, $\boldsymbol{\theta}_T$

plore dense and sparse feature encoders for obtaining the feature representation of text samples in $\mathcal{U}$. As a dense feature encoder for text samples, we use the intermediate representations as obtained from the LM that is currently being trained. We compute the representations of an input sequence by averaging the output embeddings of the constituent tokens. A question then arises on which layer of the underlying model should be used for obtaining this representation since different layers encode different types of information (Rogers et al., 2020). Another possibility is to use sparse representations such as TF-IDF (Aizawa, 2003) owing to its success at capturing statistically important lexical features (Robertson et al., 2009). We study the effect of using sparse feature representations (*i.e.*, TF-IDF) and dense feature representations obtained from different layers of LM in Section 4.3. Our experiments revealed that dense feature encoders yield the best results.

**Submodular Greedy Ordering based Data Selection:** After deciding on the choice of similarity

kernel, we now describe how to select the subsets (steps $B$ and $C$ in Figure 2) as defined by Equation (3). Approximate submodular maximization algorithms such as LazierthanLazy Greedy start with an empty subset and incrementally add data points one by one till the size of the subset equals the budget $k$ set by us. If $\mathcal{S}$ represents subset selected so far, and $e$ represents the next locally optimal data sample to be added, the submodular gain value of $e$ is defined as $f(\mathcal{S} \cup e) - f(\mathcal{S})$. While running the algorithm, we initially set the budget as the size of the entire data(say $M$) in order to obtain and store the submodular gain (step $B_2$ in Figure 2) of each data sample at the time of their addition.

The key idea here is to use the submodular gains associated with each data sample as an importance score; convert them to a probability distribution by using the second order Taylor-softmax operation (de Brébisson and Vincent, 2016) (step $C$ in Figure 2) and then sample a subset of desired size(say $k$) from the above distribution. Given gains vector $\{g_1, g_2, \cdots, g_M\}$, Taylor-softmax operation over the vector for converting it to probability distribution $P$ can be specified as $P \overset{\mathrm{def}}{=} \left\{ \frac{1+g_i+0.5g_i^2}{\sum_{j=1}^{M} 1+g_j+0.5g_j^2} \right\}_{i=1}^{M}$.

Using the probability distribution $P$ for sampling ensures that samples which have high importance score associated with them are selected with greater probability. However, it also allows the LM to explore the samples with low importance score during training to prevent overfitting. We reuse this probability distribution to sample new subsets of size $k$ every $R$ steps by sampling $k$ points without replacement (step $D$ in Figure 2).

Recall that we require a similarity kernel of size $|\mathcal{U}| \times |\mathcal{U}|$, hence the memory required for storing the similarity kernels is practically infeasible. We now describe how we scale INGENIOUS to handle size of the pre-training datasets used for LMs.

**Partitioning based Efficient Subset Selection:** To minimize the memory consumption, instead of constructing a probability distribution over the entire unlabeled set directly, we first partition (step $B_1$ in Figure 2) the unlabeled set into $N_P$ random blocks of equal sizes (*i.e.*, partition size is $\frac{|\mathcal{U}|}{N_P}$) and construct a probability distribution $P_i$ over each data block $\mathcal{U}_i^p : |\mathcal{U}_i^p| = \frac{|\mathcal{U}|}{N_P}$. We then use the con-

structed probability distributions $P_i$ over each data block $\mathcal{U}_i^p$ to sample a subset of size $k/N_P$ from the data block without replacement. We compute the final subset using subsets from each partition as follows:

$$\mathcal{S}_t = \bigcup_{i=1}^{N_P} \text{sample}\,(\mathcal{U}_i^p, P_i, \frac{k}{N_P}) \qquad (4)$$

The partitioning of the unlabeled set allows us to get away with constructing similarity kernels of size $\frac{|\mathcal{U}|}{N_P} \times \frac{|\mathcal{U}|}{N_P}$, thereby reducing the similarity kernel memory usage by around $N_P{}^2$ times. We discuss the effect of the partition size in Appendix K. In order to maximize the utilization of available resources, we can construct probability distributions over each block in the data partition in parallel. As in recent work (Mittal et al., 2022), partitioned facility location can be shown as a lower bound of the original objective function,i.e., facility location that is being maximized. It should be noted that memory utilization also increases with the number of parallel processes. For example, when $N_{PP}$ subsets are selected from partitions in parallel, the memory usage due to similarity kernel is of the order $\mathcal{O}(N_{PP}\frac{|\mathcal{U}|^2}{N_P^2})$. In our experiments, we set $N_{PP} = 100$ processes.

## 4   Experiments and Results

We use BERT (Devlin et al., 2019), GPT-2 (Radford et al., 2019) and a domain-specific version of BERT - BioBERT (Lee et al., 2020) as the underlying LMs. Specifically, we use BERT-Base(110M) and GPT2-Small(124M). For BERT, we use English Wikipedia in conjunction with BooksCorpus as the pre-training corpora and employ MLM and NSP tasks for pre-training following details in the work of Devlin et al. (2019). We perform pre-training using a batch size of 1024 for 1,000,000 steps in the case of vanilla-BERT. We perform ablations over data subset sizes and number of pre-training steps for INGENIOUS enabled pre-training and find a subset size of 25% (Appendix J) with 250,000 pre-training steps (25%) as an optimal choice. We set the value of R to 25000 steps. We refer the reader to Appendix G for further implementation details. For INGENIOUS enabled pre-training of BioBERT and GPT-2, we discuss the implementation details and experimental results in Sections 4.4 and 4.5, respectively.

|  | Avg. GLUE Score | CoLA Score |
|---|---|---|
| **Vanilla** (1M steps) | 82.72 | 55.98 |
| **Random-Selection (B1)** (250K steps) | 80.67 (-2.05%) | 51.2 (-4.78%) |
| **Early Stopping (B2)** (250K steps) | 81.23[1] (-1.49%) | 50.93 (-5.05%) |
| **Loss-based Sampling (B3)** (250K steps) | 81.25[1] (-1.47%) | 51.68 (-4.3%) |
| INGENIOUS (250K steps) | 81.57[1,2,3] (-1.15%) | 54.61[1,2,3] (-1.37%) |

Table 1: Comparison of INGENIOUS with vanilla pre-training (full 1M steps) and other baselines for BERT. We report fine-tuning performance on GLUE benchmark and CoLA task in GLUE averaged over 20 runs. Statistically significant improvements(as measured by one-tailed t-test with 99% significance level) over baselines B1, B2, B3 are indicated by superscripts 1,2,3 respectively. Numbers in brackets denote difference relative to vanilla variant. We report metrics for INGENIOUS and baselines after 250K pre-training steps. Please refer to Appendix F for task-wise scores and Appendix E for validation set losses during the course of pretraining

### 4.1   INGENIOUS for BERT Pre-training

We consider two leagues of pre-trained models, *viz.*, (i) BERT pre-trained on subsets selected through INGENIOUS and (ii) vanilla BERT pre-trained fully up to 1 million steps. We contrast these by fine-tuning each on the commonly used GLUE benchmark (Wang et al., 2019) and report the performances of each. Further, we compare INGENIOUS against three baselines - **B1) Random Selection:** which is obtained by pre-training BERT on a randomly sampled subset of the same size as that selected by INGENIOUS; **B2) Early Stopping:** BERT pre-training stopped at 250K steps as checkpoint for evaluation; **B3) Loss Based Sampling** (Loshchilov and Hutter, 2016): which is obtained by pre-training BERT on a subset, of the same size as those selected by INGENIOUS, sampled from a probability distribution that is constructed by ranking the losses in descending order and allocating the high rank (high loss) samples greater probability than low rank (low loss) samples. We would like to emphasise that we choose the baselines B1 and B3 owing to their relevance to making LM pre-training efficient with respect to data optimization. Table 1 reports the GLUE score averaged over **20 runs** on the dev sets obtained after 250K pre-training steps for the pre-trained models obtained by different methods.

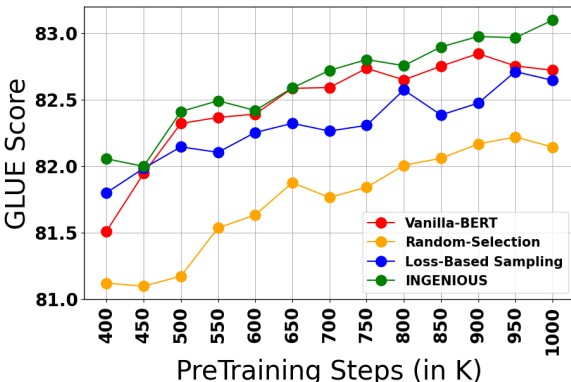

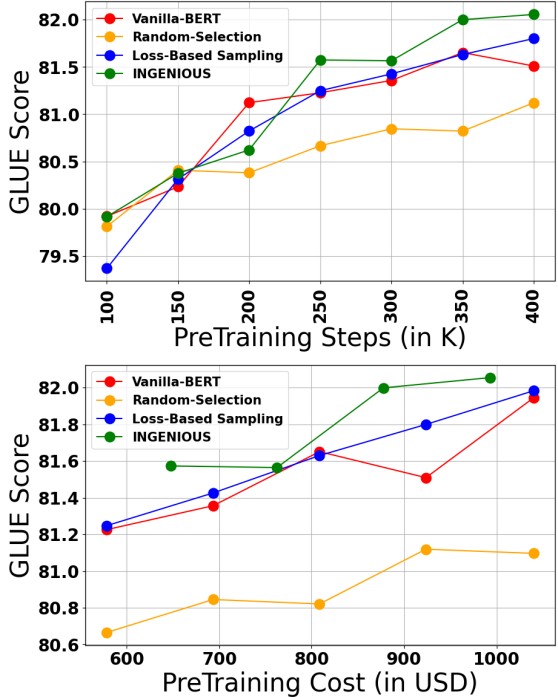

Figure 3: Comparison of INGENIOUS with vanilla BERT on GLUE performance *vs.* pre-training steps (top) and cost (bottom) using checkpoints obtained at intermediate pre-training stages.

We observe that despite using only a subset of training data and being trained only for 250K steps, INGENIOUS achieves 98.6% performance of the vanilla fully pre-trained BERT. Further, INGENIOUS achieves statistically significant improvements over the three baselines (B1, B2, and B3). INGENIOUS also outperforms baseline B3, which prioritizes training the BERT model on samples with a high loss rate. Prioritizing high-loss samples may likely result in overfitting, which may explain the poor fine-tuning performance of baseline B3 on GLUE tasks compared to baseline B2. Therefore, INGENIOUS selects informative subsets that not only help improve BERT pre-training convergence but also help retain its generalization capabilities. Further, we observe that extended training of INGENIOUS till 400K steps yields 99.1% performance of the vanilla BERT. We would like to highlight that most of the downstream task performance achieved by an PTLM is due to the initial stages of pre-training with most of the later pre-training resulting in up to $\sim 1\%$ improvement (Smith et al., 2022). In this context, INGENIOUS *helps in achieving later-stage performance gains relatively earlier*. Finally, we would like to highlight that INGENIOUS performs significantly better compared to the baselines on

Figure 4: INGENIOUS is found to outperform the baselines even on extended training till 1M steps

the CoLA task (in Table 1) which is deemed to be most difficult (Geiping and Goldstein, 2022) in the GLUE benchmark. This implies that the subsets selected by INGENIOUS are able to capture the important and highly informative signals from the underlying data resulting in robust performance on challenging tasks as well.

Further, to compare different methods at different stages of pre-training, we obtain corresponding checkpoints and fine-tune on GLUE tasks. For this particular setting, we present a comparison of vanilla BERT pre-training against INGENIOUS in Figure 3. We plot the downstream performance for all the methods and it can be seen that INGENIOUS shows better performance than all the baselines at 250K steps of pre-training and thereafter, beyond 250K steps, the trend continues consistently (Figure 3 - top). Also, pre-training through informative subsets enables BERT to achieve a performance level at 250K steps which the vanilla pre-training achieves only after over 350K iterations. Similarly, for any given pre-training cost, INGENIOUS yields a better GLUE score than the baselines (Figure 3 - bottom). Further we observe that INGENIOUS consistently outperforms the baselines even when extended to 1 million steps(maximum number of training steps prescribed by Devlin et al. (2019) for vanilla BERT pre-training) as shown in Figure 4.

**Effectiveness of Importance Sampling:** We also evaluated a variant where the samples are selected greedily based on submodular ranking instead of importance sampling over submodular gains. In contrast to the 81.57 achieved by INGENIOUS, it achieved an Avg. GLUE score of 80.5 after 250K pre-training steps, highlighting the effectiveness

| Method | Google-RE | T-REx | ConceptNet | SQuAD |
|---|---|---|---|---|
| **BERT,** **1M steps** | 3.99 | 25.76 | 11.48 | 14.77 |
| **Random Selection** **250K steps** | 4.1 | 23 | 7.87 | 11.37 |
| **BERT - Early Stopping** **250K steps** | 2.23 | 24.28 | 9.51 | 12.41 |
| **Loss-Based Sampling** **250K steps** | 2.32 | 21.66 | 7.87 | 11.76 |
| **INGENIOUS** **250K steps** | 3.39 | 24.08 | 11.15 | 13.71 |

Table 2: Knowledge retention of different models as measured by LAMA probe. We report P@1 scores for all the four different subtasks in LAMA.

| Embeddings Representation | Avg. GLUE Score |
|---|---|
| Layer 3 | 81.03 |
| Layer 6 | 80.88 |
| Layer 9 | **81.57** |
| Layer 12 | 80.93 |
| TF-IDF | 81.16 |

Table 3: Ablation study by varying embedding representation for selecting subsets. We report mean GLUE score to compare INGENIOUS variants.

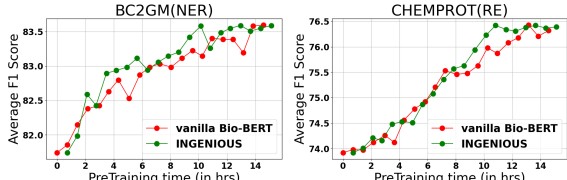

(a) BC2GM (Smith et al., 2008) Dataset   (b) ChemProt (Taboureau et al., 2011) Dataset

Figure 5: Plots (a) and (b) are the convergence results comparing Avg. F1 score (over three runs) with the Wall clock time for Vanilla BioBERT and BioBERT using INGENIOUS with 25% subset. We observe that INGENIOUS achieves much faster convergence than vanilla BioBERT(i.e., Full Training).

of importance sampling.

## 4.2 Knowledge Retention with INGENIOUS

Large PTLMs, when trained on a sufficiently large corpus, stores various types of knowledge implicitly in their parameters (AlKhamissi et al., 2022). Since INGENIOUS uses only a subset of the whole data for pre-training, it is natural for it to contain lesser knowledge in its parameters but how does it compare with vanilla BERT pretraining and other baseline when it comes to knowledge retention? To answer this question, we use LAMA benchmark (Petroni et al., 2019), a probe designed to analyze factual knowledge present in PTLMs. LAMA is derived from four distinct types of knowledge sources - Google-RE, T-REx, ConceptNet, and SQuAD – from which cloze sentences are created using facts contained in the respective knowledge sources. The PTLM has to predict the fact tokens in place of the mask tokens in cloze sentences. In Table 2, we summarize the results. We note that INGENIOUS suffers minimal loss in knowledge retention with respect to fully pre-trained vanilla BERT on all tasks. Further, the decrease in performance is less as compared to the baselines (for most tasks) which suffer a more severe decrease in performance. Intuitively, we attribute this to the ability of INGENIOUS to select highly informative subsets from the corpus while excluding the redundant information.

## 4.3 Effect of Embedding Representations

Different BERT layers have been shown to capture different information - lower layers capture word order (Rogers et al., 2020), middle capture syntactic information (Hewitt and Manning, 2019; Jawahar et al., 2019) and the later layers capture task-specific information (Kovaleva et al., 2019; Hao et al., 2019). We vary the layers - (3, 6, 9 and 12) used to obtain features for subset selection

and report the performance on GLUE in Table 3. We observe that layer 9 features yield the best results. Further, in Table 3, we compare the effect of using TF-IDF as sample representations and contrast them against dense features (BERT Layer-9). We observe that dense embeddings perform better than shallow TF-IDF features. We also report effect of subset size and number of partitions in Appendices J and K.

## 4.4 INGENIOUS for Domain-Specific PTLM - BioBERT

We evaluate the performance of Bio-BERT (Lee et al., 2020) pre-trained on subsets selected through INGENIOUS and compare it with vanilla Bio-BERT by fine-tuning it on biomedical datasets for the Named Entity Recognition (NER) and Relation Extraction (RE) tasks. For vanilla Bio-BERT, we start with a pre-trained BERT model and further pre-train it on the PubMed abstracts dataset for 200,000 steps(as recommended by the original study). Please refer to Appendix I for further implementation details. We present the performance convergence plots of vanilla Bio-BERT vs. training time using INGENIOUS with a subset size of 25% in Figure 5. It shows that during initial

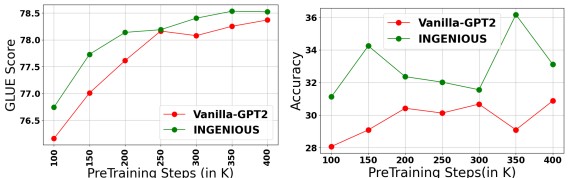

(a) GLUE (Wang et al., 2019) Benchmark  (b) BBQ Lite (Srivastava et al., 2022) Dataset

Figure 6: Comparison of INGENIOUS with vanilla GPT-2 pre-training at different pre-training stages. Pre-training on INGENIOUS subsets enables GPT-2 to achieve better GLUE score consistently.

stages of pre-training, INGENIOUS performs similar to vanilla since the LM is still learning representations, however once better representations for subset selection are learned, INGENIOUS achieves faster convergence than vanilla *w.r.t* pre-training time and achieves the best accuracy around 1.4x faster.

### 4.5 INGENIOUS for GPT-2 Pre-training

We also pre-train GPT-2 (Radford et al., 2019) using INGENIOUS. We estimate the mean accuracy for GLUE fine-tuning (averaged over 20 runs) and zero-shot accuracy on BBQ Lite generative task. Please refer to Appendix H for implementation details. We plot the performance (see Figure 6) obtained for the above benchmarks against checkpoints at different pre-training stages (steps). Figure 6 - left and right shows that INGENIOUS performs consistently better than vanilla GPT-2 pre-training on GLUE and BBQ Lite respectively at different stages of pre-training indicating better convergence.

## 5 Conclusions

We presented INGENIOUS, a framework for efficient pre-training of language models using highly informative data subsets, and presented a submodular optimization based algorithm. We described how it can be scaled for language models and showed its effectiveness using rigorous empirical evaluation. Our future work will explore exploiting external knowledge bases to identify and reduce redundancies in the corpus and to study multi-modal training where redundant information can be spread across different modalities.

## 6 Limitations

In terms of limitations, the submodular maximization based on estimation of pairwise sample sim-

ilarity can be potentially constrained by memory limitations and might require high CPU RAM capacity. Further, we do acknowledge that our experiments are performed on relatively smaller PTLMs compared to GPT-3, OPT or PaLM owing to resource limitations. We have tried our best to perform extensive experiments and perform ablation studies to inform our design choices within our resource constraints.

## 7 Ethical Considerations

We believe that INGENIOUS has a significant positive impact on society since it makes pre-training of LMs compute efficient, thereby reducing CO2 emissions and energy costs. Nonetheless, the INGENIOUS framework is susceptible to biases and toxic words within the pre-training corpora as it relies on standard pre-training datasets. An exciting future direction of this research is to investigate whether we could use targeted subset selection to filter out toxic words, as well as phrases that promote cultural stereotypes and biases from the pre-training corpora before LM pre-training.

## 8 Acknowledgments and Disclosure of Funding

This work is supported by an Adobe Data Science Award, and by the National Science Foundation under Grant No. IIS-2106937 awarded to Rishabh Iyer. Any opinions, findings, and conclusions or recommendations expressed in this material are those of the authors and do not necessarily reflect the views of Adobe or the National Science Foundation. Ganesh Ramakrishnan is grateful to the Adobe Award and Institute Chair Professorship Award at IIT Bombay for supporting this work.

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

# APPENDIX

## A  Code, Software, and Licenses

The data and code for INGENIOUS is available at the following url: https://github.com/Efficient-AI/ingenious. We release the code repositories of INGENIOUS with an MIT license, which is available for everybody to use freely.

All the code is developed using open-source HuggingFace for training LMs with PyTorch as the underlying framework. PyTorch is available under the BSD license. HuggingFace is available under Apache 2.0 license. For submodular optimization, we use a library called SUBMODLIB (Kaushal et al., 2022), which is freely available at https://github.com/decile-team/submodlib which is available under the MIT license.

## B  Additional Background and Related Work

**Submodular Functions:** Let $\mathcal{U}$ denote the *unlabeled* set of $n$ data points $\mathcal{U} = \{1, 2, 3, ..., n\}$ and a set function $f : 2^{\mathcal{U}} \to \mathbb{R}$. Formally, a function $f$ is submodular (Fujishige, 2005; Bilmes, 2022) if for $x \in \mathcal{U}$, $f(\mathcal{A} \cup x) - f(\mathcal{A}) \geq f(\mathcal{B} \cup x) - f(\mathcal{B})$, $\forall \mathcal{A} \subseteq \mathcal{B} \subseteq \mathcal{U}$ and $x \notin \mathcal{B}$. For a set $\mathcal{A} \subseteq \mathcal{U}$, $f(\mathcal{A})$ provides a real-valued score for $\mathcal{A}$. A function $f$ is said to be monotone if $f(\mathcal{A}) \leq f(\mathcal{B})$ whenever $\mathcal{A} \subseteq \mathcal{B}$. Further, f is *supermodular* if $-f$ is *submodular*, modular if it is both, and *normalized* if $f(\phi) = 0$. Submodularity occurs naturally in various real-world applications (Tohidi et al., 2020; Bach, 2013, 2019; Iyer, 2015) and a number of combinatorial functions, such as facility location, set cover, log determinant, graph cut, *etc.* (Iyer et al., 2021; Iyer and Bilmes, 2019; Kothawade et al., 2020, 2021; Karanam et al., 2022) are inherently submodular in nature. Submodularity is particularly attractive due to the constant factor $1 - \frac{1}{e}$ (Nemhauser et al., 1978) approximation for cardinality-constrained submodular maximization, allowing us to solve various combinatorial optimization problems, which are often NP-Hard in nature. Several recent works (Wei et al., 2014a, 2015; Mirzasoleiman et al., 2020; Killamsetty et al., 2021b,a,c, 2022; Kothawade et al., 2021; Karanam et al., 2022) have formulated the subset selection objective as a submodular maximization problem. Furthermore, variants of the greedy algorithm (Mirzasoleiman et al., 2015; Iyer and Bilmes, 2019) that can maximize a submod-

ular function in *near-linear time* have been proposed.

**Submodular Data Subset Selection:** Submodular optimization has been successfully employed for data subset selection in various applications such as speech recognition (Wei et al., 2014b,a; Mittal et al., 2022), machine translation (Kirchhoff and Bilmes, 2014), active-learning (Wei et al., 2015; Kothawade et al., 2021), efficient deep learning (Kaushal et al., 2019; Killamsetty et al., 2022; Pooladzandi et al., 2022). Another active area of research is selecting representative subsets of data, also known as *coresets* (Feldman, 2020). A coreset is a weighted subset of data closely approximating certain desirable properties of the entire dataset (e.g., the loss function) (Feldman, 2020). Coreset selection has been shown to benefit a host of geometric problems such as $k$-means and $k$-median clustering (Har-Peled and Mazumdar, 2004) and, in recent times, has been used successfully for efficient bayesian inference (Campbell and Broderick, 2018) and improving training efficiency (Mirzasoleiman et al., 2020; Killamsetty et al., 2021a). Such informative data subset selection has shown remarkable promise for efficient and robust training of deep models (Killamsetty et al., 2021b,c). We direct the reader to a survey by Bilmes (2022) for a detailed review of submodularity and subset selection for ML.

## C   Datasets

For pre-training BERT, we use English Wikipedia, BooksCorpus datasets. English Wikipedia is ~20GiB of text containing 6,458,670 articles and BooksCorpus is ~5GiB of text containing 74,004,228 lines of text. For GPT-2, we use Open-WebText which is an open source replication of WebText dataset from OpenAI. OpenWebtext is around ~40GiB of text around 8,013,769 articles.

## D   Compute Infrastructure

All our pre-trainings were done on Google Cloud Platform (GCP) instances comprising of 8 NVIDIA A100-SXM4-40GB GPUs for BERT(bert-base-uncased: 110M parameters) and 16 NVIDIA A100-SXM4-40GB GPUs for GPT-2(gpt2-small: 124M parameters). In each instance, there are 96 CPU cores with a total RAM of 680GiB. The costs are estimated using https://cloud.google.com/products/calculator based on the time taken for training.

## E   Pre-Training performance of INGENIOUS for BERT

In Table 7, we show how validation-set losses change for vanilla BERT and INGENIOUS BERT over the course of pretraining.

## F   GLUE Task wise performance of INGENIOUS for BERT

We show task-wise performance on GLUE for BERT trained through INGENIOUS in Table 4. We compare against vanilla LM pre-training and baselines. We also report the standard deviation for each task along with the mean glue score.

## G   Further implementation details of pre-training BERT through INGENIOUS

We use Adam optimizer (Kingma and Ba, 2014) with learning rate of 1e-4, $\beta_1 = 0.9$, $\beta_2 = 0.99$, L2 weight decay of 0.01. We warmstart the model for the first 80K training steps and subsequently train only on selected subsets. Training of all the models is performed on 8 NVIDIA A100-SXM4-40GB GPUs.

## H   Implementation details of pre-training GPT-2 through INGENIOUS

For GPT-2, we use OpenWebtext(An open-source replication of WebText dataset from OpenAI) as the pre-training corpus and employ CLM task for pre-training following details in the work of Radford et al. (2019). We perform pre-training using a batch size of 256 (achieved using gradient accumulation of 2 steps) for 1,000,000 steps in case of vanilla GPT-2. With INGENIOUS, we pre-train GPT-2 for 250,000 steps. We set the value of R as 25K steps. We use Adam Optimizer (Kingma and Ba, 2014) with learning rate of 1e-4, $\beta_1 = 0.9$, $\beta_2 = 0.99$, L2 weight decay of 0.01. We warmstart the model the first 65K training steps and subsequently train only on selected subsets. Training of all the models is performed on 16 NVIDIA A100-SXM4-40GB GPUs.

## I   Implementation details of pre-training BioBERT through INGENIOUS

We use Adam optimizer (Kingma and Ba, 2014) with learning rate of 1e-4, $\beta_1 = 0.9$, $\beta_2 = 0.99$, L2 weight decay of 0.01. For Bio-BERT training using INGENIOUS, we use a subset size of 25% ,

| Method | Pre-Training Cost (USD) | Mean Avg. GLUE Score | CoLA | MRPC | RTE | STS-B | SST-2 | MNLI (matched) | MNLI (mismatched) | QNLI | QQP |
|---|---|---|---|---|---|---|---|---|---|---|---|
| **Vanilla BERT, 1M steps** | $2,309.53 | $82.72_{\pm0.35}$ | $55.98_{\pm1.93}$ | $90.55_{\pm0.95}$ | $69.31_{\pm2.05}$ | $89.25_{\pm0.24}$ | $92.08_{\pm0.52}$ | $83.89_{\pm0.27}$ | $84.26_{\pm0.22}$ | $91.26_{\pm0.24}$ | $87.9_{\pm0.14}$ |
| **Random-Selection, 250K steps** | $578.44 (-74.95%) | $80.67_{\pm0.27}$ (-2.05) | $51.2_{\pm1.77}$ | $89.25_{\pm0.77}$ | $65.42_{\pm1.7}$ | $87.57_{\pm0.32}$ | $90.68_{\pm0.64}$ | $82.05_{\pm0.23}$ | $82.76_{\pm0.17}$ | $89.69_{\pm0.27}$ | $87.37_{\pm0.15}$ |
| **Vanilla BERT Early stopping, 250K steps** | $578.44 (-74.95%) | $81.23_{\pm0.34}$ (-1.49) | $50.93_{\pm2.3}$ | $90.57_{\pm0.89}$ | $66.26_{\pm1.47}$ | $88.89_{\pm0.34}$ | $90.9_{\pm0.46}$ | $82.56_{\pm0.26}$ | $83.1_{\pm0.18}$ | $90.23_{\pm0.2}$ | $87.58_{\pm0.13}$ |
| **Loss-Based Sampling, 250K steps** | $578.44 (-74.95%) | $81.25_{\pm0.26}$ (-1.47) | $51.86_{\pm1.99}$ | $89.83_{\pm0.78}$ | $66.82_{\pm1.55}$ | $88.5_{\pm0.25}$ | $90.63_{\pm0.53}$ | $82.57_{\pm0.17}$ | $83.21_{\pm0.19}$ | $90.17_{\pm0.25}$ | $87.65_{\pm0.13}$ |
| **INGENIOUS, 250K steps** | $647.56 (-71.96%) | $81.57_{\pm0.37}$ (-1.15) | $54.61_{\pm1.64}$ | $89.68_{\pm0.83}$ | $67.16_{\pm1.64}$ | $88.94_{\pm0.21}$ | $91.01_{\pm0.5}$ | $82.15_{\pm0.22}$ | $82.84_{\pm0.23}$ | $90.25_{\pm0.2}$ | $87.53_{\pm0.15}$ |

Table 4: Comparison of pre-training cost and fine-tuning performance on GLUE tasks (averaged over 20 runs) for BERT. We report difference relative to full pre-training of vanilla BERT in brackets for cost and avg. GLUE score. INGENIOUS achieves 98.6% of fully pre-trained BERT performance, reducing pre-training cost to $\sim 28\%$.

| Subset Size (% of full dataset) | Avg. GLUE Score |
|---|---|
| **10%** | 81.18 |
| **15%** | 81.24 |
| **20%** | 81.10 |
| **25%** | **81.57** |
| **30%** | 81.17 |

Table 5: Ablation study by varying subset size of selected subsets. We report mean GLUE score to compare INGENIOUS variants.

| Number of partitions | Avg. GLUE Score |
|---|---|
| **1500** | **81.57** |
| **2000** | 81.19 |
| **2500** | 81.37 |
| **3000** | 81.34 |

Table 6: Ablation study by varying partition size used during pre-training. We report mean GLUE score to compare INGENIOUS variants.

$R$ value of 5000, no model warm-start, i.e., $W = 0$, and trained the Bio-BERT model for 200,000 steps.

## J   Subset size for efficiency gains

We study the effect of the size of the subset selected through INGENIOUS that is used for pre-training BERT. In Table 5, we analyse using the following values of subset sizes, *viz.*, 10%, 15%, 20%, 25% and 30% and evaluate the fine-tuning performance on GLUE. While lower subset sizes (10-20%) result in inferior performance owing to the fact that the LM is shown less information, optimal performance is observed when 25% of the pre-training corpus is used, hence, we report corresponding results in Table 1.

## K   Partitions for efficient subset selection

As discussed in approach, we divide the pre-training dataset into partitions. In Table 6, we analyse the impact of performance on GLUE as the number of partitions is varied. Using fewest partitions (1500) is found to yield optimal performance. This aligns with the intuition that fewest partitions enable better subset selection since more samples are present in a single partition, allowing to select more representative samples overall.

## L   Few Examples of informative texts sampled by INGENIOUS

We summarize the three types of redundancies that we found in our analysis of selected subsets. More examples can be found at https://github.com/Effic ient-AI/ingenious.

- **Type 1:** Same information conveyed by multiple sentences in different documents.
  - **Sentence 1:** "separate sovereign countries but acted as a single bloc in foreign policy and security issues. the proposed union was being discussed by a joint scandinavian committee during the winter of 1948 – 1949, but the cold war tension between the united states and the soviet union, and preparations for a western alliance that would result in the north atlantic treaty overshadowed the effort. when it became"
  - **Sentence 2:** "they would remain separate sovereign countries but act as a single block in foreign policy and security issues. the proposed union was discussed by a joint scandinavian committee during the winter of 1948 – 1949, but in the end the cold war tension between the united states and the soviet union and preparations for a western alliance that would result in"

| Pre-Training Steps | 100K | 150K | 200K | 250K | 300K | 350K | 400K | 450K | 500K | 550K | 600K | 650K | 700K | 750K | 800K | 850K | 900K | 950K | 1000K |
|---|---|---|---|---|---|---|---|---|---|---|---|---|---|---|---|---|---|---|---|
| Vanilla BERT | 2.29 | 2.07 | 1.99 | 1.92 | 1.88 | 1.86 | 1.83 | 1.8 | 1.79 | 1.78 | 1.76 | 1.74 | 1.73 | 1.73 | 1.71 | 1.7 | 1.69 | 1.69 | 1.68 |
| INGENIOUS BERT | 2.29 | 2.09 | 2.01 | 1.95 | 1.91 | 1.88 | 1.85 | 1.85 | 1.82 | 1.8 | 1.79 | 1.78 | 1.76 | 1.76 | 1.74 | 1.74 | 1.72 | 1.71 | 1.7 |

Table 7: Comparison of validation set losses during pre-training. INGENIOUS achieves almost similar validation set loss as compared to vanilla BERT

- **Type 2:** Duplicates in the corpus.
    - **Sentence 1:** "after we'd been handed our menus. i always get the frozen hot chocolate." frozen hot chocolate? it's really a thing? i thought they just made that up." no," i said, pointing to the spot on her menu. see? it's right there." so, do you order anything else?" cake." she looked at my deadpanned face and laughed. so, we're"
    - **Sentence 2:** "what's good here?" mia asked me after we'd been handed our menus. i always get the frozen hot chocolate." frozen hot chocolate? it's really a thing? i thought they just made that up." no," i said, pointing to the spot on her menu. see? it's right there."
- **Type 3:** Recurring patterns of text.
    - **Sentence 1:** "according to the united states census bureau, the village has a total area of, all land. demographics 2010 census as of the census of 2010, there were 377 people, 159 households, and 101 families residing in the village. the population density was. there were 176 housing units at"
    - **Sentence 2:** "according to the united states census bureau, the village has a total area of, all land. demographics 2010 census as of the census of 2010, there were 801 people, 323 households, and 225 families living in the village. the population density was. there were 358 housing units at"