# OpenReview forum: "INGENIOUS: Using Informative Data Subsets for Efficient Pre-Training of Language Models"
_EMNLP/2023/Conference — EMNLP 2023 Findings_

### Official Review · Reviewer_NYV4 · 2023-08-05

**Soundness:** 2

**Excitement:**

3: Ambivalent: It has merits (e.g., it reports state-of-the-art results, the idea is nice), but there are key weaknesses (e.g., it describes incremental work), and it can significantly benefit from another round of revision. However, I won't object to accepting it if my co-reviewers champion it.

**Paper Topic And Main Contributions:**

This paper proposes a framework for improving the training efficiency of pre-trained language models (PTLMs) by selecting highly informative subsets of the training data. The framework employs submodular optimization to select representative subsets of the training corpora and can be applied to efficiently train multiple PTLMs using only a fraction of data. The resulting models achieve up to 90% of the performance of fully-trained models. The contribution of this paper lies in the development of a new training framework that significantly improves the training efficiency of PTLMs while maintaining downstream performance.

**Reasons To Accept:**

This paper investigates the issue of resource consumption in training large language models (LLMs) by sampling high-quality information from the training dataset to ensure efficient learning of effective information. This topic is crucial for the efficient training of LLMs. The article provides a clear training strategy and validates its effectiveness on multiple models (but not large enough), which is inspiring.

**Reasons To Reject:**

While this paper's research is insightful, it has some limitations. The current issues with training data quality for LLMs go beyond efficiency and information quantity, including concerns about values, toxicity, and biases. Additionally, the paper's validation was limited to BERT and GPT2-small models and evaluated using GLUE metrics, which may not be sufficient to support the conclusions.

**Reproducibility:**

5: Could easily reproduce the results.

**Reviewer Confidence:**

2: Willing to defend my evaluation, but it is fairly likely that I missed some details, didn't understand some central points, or can't be sure about the novelty of the work.

---

> ### Author Rebuttal · Authors · 2023-08-29
>
> Thank you for appreciating our proposed training strategy for the crucial task of efficient LM training.
>
> We believe there has been a misunderstanding regarding the fundamental argument of the paper. Our key goal is **efficient** training of the LMs. Our proposal is to identify and remove redundancies in the training data. And our rigorous experiments show that handling redundancies and noise and feeding high-quality samples to the model results in faster learning. While bias, toxicity, and values are significant factors to consider for LM training, they are orthogonal to the focus of current work -- efficient training. Our focus is not on measuring or creating high-quality datasets for training.
>
> Further, our evaluations are not just limited to the GLUE benchmark. While GLUE is one of the most commonly used benchmarks comprising multiple tasks for LM evaluation, we go beyond that and evaluate the knowledge retention capabilities by the LAMA benchmark (Section 4.2), study the performance for domain-specific applications (Section 4.4), perform thorough ablation studies (Section 4.1, 4.3), and use different types of LMs (Section 4.5). Moreover, note that we report the numbers averaged over 20 runs to account for the effects of any random seed. Very rarely, if ever, such stringent evaluation is reported for LM training.
>
> Regarding the model size, as already acknowledged in the "Limitations" section of our paper, we used BERT and GPT-2 small models due to resource constraints. However, do note that we were training these models from scratch and not just fine-tuning. To account for our lack of financial and compute resources (to be honest, very few academic institutions, if any, can afford to train large models from scratch), we ensured a rigorous evaluation scheme to confirm the validity and efficacy of our approach, including thorough ablation studies and multiple runs (as also noted by all the other reviewers).
>
> We have made all our implementations open and free for others to build upon, and we remain confident that the performance enhancements observed in our approach will similarly manifest in larger models.

---

### Official Review · Reviewer_m6xK · 2023-08-05

**Soundness:** 2

**Excitement:**

3: Ambivalent: It has merits (e.g., it reports state-of-the-art results, the idea is nice), but there are key weaknesses (e.g., it describes incremental work), and it can significantly benefit from another round of revision. However, I won't object to accepting it if my co-reviewers champion it.

**Paper Topic And Main Contributions:**

This paper addresses the problem of a large computational cost of pre-training language models due to the size of pre-training corpora. To reduce the pre-training cost, the authors propose to extract a highly informative subset of pre-training corpus. The experimental results in the paper show that the proposed method outperforms early stopping and random extraction methods in terms of fine-tuned GLUE scores.

**Questions For The Authors:**

A. Would you provide some examples of informative texts sampled by the proposed method? What differences do exist between the informative texts and the others?

B. In Sec. 4.4, the proposed method achieves final performance on par with vanilla BERT. What does happen if a model is trained for longer epochs? It seems that the performance of the proposed method converges while vanilla BERT keeps improving.

C. The paper reported that 25% of training corpus is optimal, outperforming vanilla BERT when fully trained. Would you explain why 75% of training corpus deteriorates the performance of language models?


**Reasons To Accept:**

A. The paper addresses an important problem of the pre-training cost of language models for efficient NLP.

B. The paper presents a novel approach to remove redundant texts in pre-training corpora in NLP domains.

C. Experimental results that language models trained with informative subsets outperform vanilla models in terms of convergence speed and final performance.

**Reasons To Reject:**

A. Although extracting data subsets work well in computer vision tasks, texts have different properties compared with images in nature. The paper focuses on the size of datasets and lacks discussion about the different modality.

B. Most experiments are conducted on models with ~110M parameters. It is unclear whether the proposed method consistently improves the efficiency with scaled models.

C. Lack of analysis and discussion leaves several questions about the reasons of performance improvements.


**Reproducibility:**

4: Could mostly reproduce the results, but there may be some variation because of sample variance or minor variations in their interpretation of the protocol or method.

**Reviewer Confidence:**

4: Quite sure. I tried to check the important points carefully. It's unlikely, though conceivable, that I missed something that should affect my ratings.

---

> ### Author Rebuttal · Authors · 2023-08-29
>
> Thank you for your encouraging remarks and for appreciating the importance of the problem and the experimental rigor. We appreciate the inquisitive questions which we answer below.
>
> **Determining Useful Information for Text Datasets:**
>
> >Although extracting data subsets work well in computer vision tasks, texts have different properties compared with images in nature. The paper focuses on the size of datasets and lacks discussion about the different modality.
>
> Exactly! Text and image data have very different characteristics and this is precisely our argument in the paper that approaches used in Vision have to be adapted to make them work for text data. We focused on selecting smaller subsets for efficiency and also presented discussion in Section 4.3 that delves into details of text-specific characteristics that help us adapt subset selection for LM training. Specifically,  our findings indicate that the ninth layer of BERT-base excels in selecting informative subsets crucial for model pre-training (Table 3)and is more advantageous than initial or later layers. In Section (4.3), we discuss these findings in light of the seminal study[1], which showed that the middle layers (6-9) of BERT predominantly capture syntactic nuances, with layers 8 and 9 showing enhanced subject-verb agreement. Further, the initial layers, like layer 3, focus on word order, while the later layers are tailored to capture the specifics of downstream tasks. It is also worth noting that semantic information is permeated throughout the network. Thus, we present a thorough ablation of what characteristics of language data are beneficial for the subset selection task in contrast to the image data.
>
> **Size of the models:**
>
> >Most experiments are conducted on models with ~110M parameters. It is unclear whether the proposed method consistently improves the efficiency with scaled models.
>
> As acknowledged in the Limitations section of the paper, we do note that our experiments are performed on moderate-sized language models owing to resource limitations. However, do note that we trained these models from scratch and not just fine-tuned. Given the scale and rigor of our experiments (multiple settings, ablations, and multiple runs for each experiment), even training the three models used in the paper is a significant endeavor. To account for our lack of financial and compute resources (to be honest, very few institutions, except for the largest mega corporations, can afford to train large models from scratch), we ensured a rigorous evaluation scheme to confirm the validity and efficacy of our approach, including thorough ablation studies and multiple runs (20 times).
>
> **Examples of text selected in the subsets:**
>
> >Would you provide some examples of informative texts sampled by the proposed method? What differences do exist between the informative texts and the others?
>
> Great question! We summarize the three types of redundancies that we found in our analysis of selected subsets here, and will be adding these to the Appendix as well.
>
> **Type-I:** We discovered cases in which the exact same information is conveyed by multiple sentences in different documents, in which case only one of the sentences is assigned a high score to be included in the training sample.
> Example:
>
> Sentence 1: "separate sovereign countries but acted as a single bloc in foreign policy and security issues. the proposed union was being discussed by a joint scandinavian committee during the winter of 1948 – 1949, but the cold war tension between the united states and the soviet union, and preparations for a western alliance that would result in the north atlantic treaty overshadowed the effort. when it became"
>
> Sentence 2: "they would remain separate sovereign countries but act as a single block in foreign policy and security issues. the proposed union was discussed by a joint scandinavian committee during the winter of 1948 – 1949, but in the end the cold war tension between the united states and the soviet union and preparations for a western alliance that would result in"
>
> **Type-II:** We discovered many cases (mostly in the Bookcorpus dataset) where there are duplicates in the corpus, in which case only one of the sentence is assigned a high score for inclusion.
>
> Example:
>
> Sentence-1: “after we'd been handed our menus. ` ` i always get the frozen hot chocolate.'' ` ` frozen hot chocolate? it's really a thing? i thought they just made that up.'' ` ` no,'' i said, pointing to the spot on her menu. ` ` see? it's right there.'' ` ` so, do you order anything else?'' ` ` cake.'' she looked at my deadpanned face and laughed. ` ` so, we're”
>
> Sentence-2: “what's good here?'' mia asked me after we'd been handed our menus. ` ` i always get the frozen hot chocolate.'' ` ` frozen hot chocolate? it's really a thing? i thought they just made that up.'' ` ` no,'' i said, pointing to the spot on her menu. ` ` see? it's right there.”
>
> **Type-III:** We discovered a few recurring patterns of templates in the corpus, in which case INGENIOUS assigns a high score to an example e1 of the template T and a low score to other examples e2, e3,... of the same template T. We posit that this way the redundancy in language is being minimized.
>
> Example:
>
> Sentence-1: “according to the united states census bureau, the village has a total area of, all land. demographics 2010 census as of the census of 2010, there were 377 people, 159 households, and 101 families residing in the village. the population density was. there were 176 housing units at”
>
> Sentence-2: “according to the united states census bureau, the village has a total area of, all land. demographics 2010 census as of the census of 2010, there were 801 people, 323 households, and 225 families living in the village. the population density was. there were 358 housing units at”
>
> We will also release the complete pre-processed dataset used and the selected subsets and add many more examples like the above in the Appendix.
>
> **Convergence of Bio-BERT in Tandem with INGENIOUS:**
>
> >In Sec. 4.4, the proposed method achieves final performance on par with vanilla BERT. What does happen if a model is trained for longer epochs? It seems that the performance of the proposed method converges while vanilla BERT keeps improving.
>
> In Section 4.4, we trained both the standard and the INGENIOUS versions for the full number of epochs as recommended by the original study. INGENIOUS displayed a **faster** rate of convergence, achieving the performance of the fully-trained standard model in approximately **10** hours, compared to the latter's 14 hours. This is the key insight from the result of this experiment. Although the vanilla Bio-BERT model seems not to have converged yet, we expect that with extended training, it too will stabilize, as suggested by the trends observed in its last three checkpoints. We plan to incorporate further experiments with model training for a greater number of steps in the next version of the paper.
>
> **Subset Size Explanation in INGENIOUS:**
>
> >The paper reported that 25% of training corpus is optimal, outperforming vanilla BERT when fully trained. Would you explain why 75% of training corpus deteriorates the performance of language models?
>
> A subtle point to note here. In INGENIOUS, when we mention selecting a 25% subset, it doesn't imply that only 25% of the dataset is used for model training. As illustrated in Figure 2 of the paper and in the pseudo-code in the Appendix C, after every 25K steps, we consistently sample a fresh subset that constitutes 25% of the data, based on the updated probability distribution. This method ensures that the model consistently engages with nearly the entire dataset over time. Moreover, the learned probability distribution prioritizes data points deemed more informative through submodular functions. As a result, more informative data points are sampled more often, while less significant points are sampled less frequently. We argue that in contrast to INGENIOUS, with no subset selection in place in the Vanilla BERT training, the model spends much time on less informative and noisy data points, leading to sub-optimal performance.
>
> [1] Rogers, A., Kovaleva, O. and Rumshisky, A., 2021. A primer in BERTology: What we know about how BERT works. Transactions of the Association for Computational Linguistics, 8, pp.842-866.

---

### Official Review · Reviewer_Ex9w · 2023-08-09

**Typos Grammar Style And Presentation Improvements:** NA
**Soundness:** 2

**Excitement:**

3: Ambivalent: It has merits (e.g., it reports state-of-the-art results, the idea is nice), but there are key weaknesses (e.g., it describes incremental work), and it can significantly benefit from another round of revision. However, I won't object to accepting it if my co-reviewers champion it.

**Missing References:**

NA

**Paper Topic And Main Contributions:**

This paper proposes INGENIOUS, a framework for iteratively selecting informative subsets of the pre-training corpus by optimizing a submodular maximization problem wherein a similarity-based facility location acts as the submodular function. By pursuing this objective, the framework identifies a collection of diverse samples, each representing similar samples within the corpus.

The authors empirically validate the approach by pre-training language models on the selected subsets. These models are fine-tuned across various glue downstream tasks, which yield performance comparable to models that were pre-trained on the entire dataset and fine-tuned for the same tasks. The INGENIOUS-pretrained models display increased efficiency, achieving the later-stage performance of baseline models at an earlier time.


**Questions For The Authors:**

Can you provide a more detailed explanation of how figures 3 and 4 were generated? How exactly are the values in y-axis computed? Additionally, can you clarify which specific glue task was used in these figures?

Do you have any insights into why training on selected data can yield better results compared to a fully pre-trained model? Are there factors beyond noise reduction?

Can you share a pre-training performance comparison between fully pre-trained BERT (e.g., Bert-base) and INGENIOUS pre-trained BERT, without downstream fine-tuning? One suggestion could be something like testing accuracy/loss on the original validation/test set.

Will it be possible for you to share a comparison of the pre-training performance between a fully pre-trained BERT (such as Bert-base) and an INGENIOUS pre-trained BERT, without any downstream fine-tuning? One suggestion on evaluation metrics could be testing accuracy or loss on the original validation/test set.

Will you release the selected subsets that yield your best models?

I will raise my scores if the above concerns can be addressed/clarified by the rebuttal.

**Reasons To Accept:**

Firstly, the proposed method, INGENIOUS, demonstrates an impressive performance. After fine-tuning, the model pre-trained using INGENIOUS-selected subsets achieves nearly identical performance compared to models pre-trained with the complete dataset, and even surpasses them with reduced training costs (as seen in Table 3).

Then, the experimental design is comprehensive, covering comparisons across various architectures, previous sampling methods, and different downstream tasks. The evaluation is also multi-faceted, assessing both performance and cost considerations.

Additionally, the paper maintains a well-organized structure. The experiment and analysis sections are good in depth and detail, though the methodology section could benefit from a more detailed elaboration.

**Reasons To Reject:**

First, I’m confused by Figures 3 and 4. In these illustrations, INGENIOUS appears to consistently and significantly outperform the baseline BERT model which undergoes full pre-training, even when it is trained up to 1 million pre-training steps. However, based on the context along with Tables 1&2 and the detailed Table 6 in Appendix F, it becomes evident that the most proficient model pre-trained with INGENIOUS achieves a 98.6% performance compared to the fully pre-trained BERT. This contrast in results is confusing. And if Figures 3&4 is sensible, what is the rationale behind the phenomenon that pre-training on selected subsets can surpass the performance of a fully pre-trained model? Does this suggest that the original pre-training corpus contains harmful noise that interferes with learning?

Secondly,  for a thorough evaluation of INGENIOUS' effectiveness in selecting valuable subsets for pre-training, it is necessary to see a comparison between the pre-training performance of a model using INGENIOUS-selected data and one using the entire dataset, before any downstream fine-tuning.  The clean pre-training performance gains of INGENIOUS-selected data can demonstrate the evaluation process better.


**Reproducibility:**

3: Could reproduce the results with some difficulty. The settings of parameters are underspecified or subjectively determined; the training/evaluation data are not widely available.

**Reviewer Confidence:**

3: Pretty sure, but there's a chance I missed something. Although I have a good feel for this area in general, I did not carefully check the paper's details, e.g., the math, experimental design, or novelty.

---

> ### Author Rebuttal · Authors · 2023-08-29
>
> Sincere thanks for the very detailed review and appreciating the proposed approach, writing and thorough experiments. Here are tthe detailed response for the clarifications sought.
>
> **Clarification on Figures (3,4) and Tables (1,2,6):**
>
> >First, I’m confused by Figures 3 and 4. In these illustrations, INGENIOUS appears to consistently and significantly outperform the baseline BERT model which undergoes full pre-training, even when it is trained up to 1 million pre-training steps. However, based on the context along with Tables 1&2 and the detailed Table 6 in Appendix F, it becomes evident that the most proficient model pre-trained with INGENIOUS achieves a 98.6% performance compared to the fully pre-trained BERT. This contrast in results is confusing.
>
> In the Tables 1, 2 and 6, when we say INGENIOUS attains 98.6% of vanilla-BERT’s performance, we intended to convey that INGENIOUS trained for **just 250K steps** is able to attain 98.6% of the performance of vanilla BERT that is pre-trained till **1M steps**. Thus implying that INGENIOUS can lead to significantly faster learning. While Figures 3 (top) and 4 bothconvey the same point, there is a subtle difference.  Here,  we present the performance (measured by GLUE score) achieved by different training methods for the **same** number of pre-training steps (on the x-axis). We see that INGENIOUS achieves higher downstream performance than the vanilla BERT trained for the same number of steps, again implying faster learning given a compute budget.
>
>
>
> **Reasons for INGENIOUS Outperforming Vanilla-BERT:**
> >Do you have any insights into why training on selected data can yield better results compared to a fully pre-trained model? Are there factors beyond noise reduction?
> Recall from Section 3.4 that INGENIOUS employs a probability distribution constructed through submodular gains to sample subsets. This method prioritizes data points with higher importance scores i.e., that are more informative and are thus, more likely to be included in the subset used for training. As a result, the model frequently encounters these high-importance samples, while reducing exposure to redundant data points. We believe this selective exposure to more informative samples is the primary reason INGENIOUS excels. The model achieves better performance than vanilla BERT by effectively reducing redundancy in the training corpus, as INGENIOUS does. The enhancement in LLM performance due to the reduction of redundancy is also supported by recent studies [1] where authors demonstrated that de-duplicating training data significantly improves language model training.
>
>
> **Y-axis in Figures 3 and 4:**
>
> >Can you provide a more detailed explanation of how figures 3 and 4 were generated? How exactly are the values in y-axis computed? Additionally, can you clarify which specific glue task was used in these figures?
>
> Sure. As discussed in the paper (Section 4.1), the y-axis in Figures 3 and 4 is the average GLUE score (averaged over 20 fine-tuning runs). In each run, we compute the GLUE score as the average performance across the nine GLUE tasks - CoLA, MRPC, STS-B, RTE, SST-2, MNLI(m), MNLI(mm), QNLI, QQP.
>
> **INGENIOUS vs vanilla-BERT(validation loss comparison):**
>
> >Can you share a pre-training performance comparison between fully pre-trained BERT (e.g., Bert-base) and INGENIOUS pre-trained BERT, without downstream fine-tuning? One suggestion could be something like testing accuracy/loss on the original validation/test set.
>
> Great suggestion! Following is a table with validation set losses of INGENIOUS and vanilla-BERT during the course of pre-training:
>
>
>
> | Training Steps | 100K | 150K | 200K | 250K | 300K | 350K | 400K | 450K | 500K | 550K | 600K | 650K | 700K | 750K | 800K | 850K | 900K | 950K | 1000K |
> | ---- | ----| ----| ----| ----| ----| ----| ----| ----| ----| ----| ----| ----| ----| ----| ----| ----| ----| ----| ----|
> | vanilla BERT | 2.29 | 2.07 | 1.99 | 1.92 | 1.88 | 1.86 | 1.83 | 1.8 | 1.79 | 1.78 | 1.76 | 1.74 | 1.73 | 1.73 | 1.71 | 1.7 | 1.69 | 1.69 | 1.68 | 1.68 |
> | INGENIOUS BERT | 2.29 | 2.09 | 2.01 | 1.95 | 1.91 | 1.88 | 1.85 | 1.85 | 1.82 | 1.8 | 1.79 | 1.78 | 1.76 | 1.76 | 1.74 | 1.74 | 1.72 | 1.71 | 1.7 | 1.7 |
>
> As can be seen from the above table, INGENIOUS achieves almost similar validation loss as compared to vanilla BERT.
>
>
> **Releasing the subsets selected by INGENIOUS**
>
> >Will you release the selected subsets that yield your best models?
>
> Yes, definitely! Our code is already public on anonymous github. We will release the pre-processed dataset that is used and the selected subsets in the form of a Google Drive link due to the size. (Not releasing the link now to respect the anonymity guidelines)
>
> [1] Lee, K., Ippolito, D., Nystrom, A., Zhang, C., Eck, D., Callison-Burch, C. and Carlini, N., 2021. Deduplicating training data makes language models better. arXiv preprint arXiv:2107.06499.

---

### Official Review · Reviewer_ETAv · 2023-08-12

**Soundness:** 4

**Excitement:**

4: Strong: This paper deepens the understanding of some phenomenon or lowers the barriers to an existing research direction.

**Paper Topic And Main Contributions:**

The main contribution of this paper is a method to choose a subset of the pretraining model data that achieves similar performance as training on the full dataset. Data pruning methods have been extensively studied in the literature, but relatively little is known about optimal strategies for language model pretraining. This work presents an important and interpretable method to do so, and quantifies the trade-offs between cost and model performance.

**Reasons To Accept:**

Overall, the paper is well written. The experiments are thorough, and I particularly appreciate the conversion of performance metrics to costs, which lets practitioners more easily evaluate trade-offs.

**Reasons To Reject:**

There are 3 potential changes that would improve this work:
* First, something that didn't come across was the importance and intuition behind the choice of the similarity kernel. What types of kernels work best? Are there, e.g., cheap empirical metrics that can effectively estimate the clustering kernel in eq. 2? Could you estimate this similarity kernel through something very simple, such as SentenceBERT embeddings? What features is it capturing that makes it particularly good?
* Including a comparison to one of the methods mentioned in the computer vision setting would have been more useful than comparing to, e.g. loss-based sampling. I understand that these are not always applicable and typically require a supervised set-up, but some of them can probably be adapted to language tasks relatively easily.
* Not sure I understand why increasing the subset size in some cases actually hurts performance? e.g., Table 4. I think the paper could benefit by elaborating on this, and why some subsets of the dataset are actually harmful to model performance.

--

Edit: I acknowledge the authors' comments below. Although I still have some questions/concerns about the subset ablations (and hope that the authors will eventually include a more granular analysis of why this happens, and if decoder only models exhibit some type of this behavior), I am moving to increase my soundness score as they have addressed my other comments.

**Reproducibility:**

4: Could mostly reproduce the results, but there may be some variation because of sample variance or minor variations in their interpretation of the protocol or method.

**Reviewer Confidence:**

4: Quite sure. I tried to check the important points carefully. It's unlikely, though conceivable, that I missed something that should affect my ratings.

---

> ### Author Rebuttal · Authors · 2023-08-29
>
> Thank you for appreciating our work, the writing and the thorough evaluations. Your constructive feedback is highly appreciated and here are our detailed responses for your suggestions.
>
> **Choice of similarity kernel in Eq. (2):**
>
> > First, something that didn't come across was the importance and intuition behind the choice of the similarity kernel. What types of kernels work best? Are there, e.g., cheap empirical metrics that can effectively estimate the clustering kernel in eq. 2? Could you estimate this similarity kernel through something very simple, such as SentenceBERT embeddings? What features is it capturing that makes it particularly good?
>
> The similarity matrix in Eq. (2) is vital as the pairwise similarity values help capture "redundancy". One of the most common metrics for computing similarity between two pieces of text is the cosine similarity between their vector representations. The critical decision, thus, is to decide on the choice of vector representations. As you noted, one alternative is to use embeddings from pre-trained models (e.g, SentenceBERT). However, note that we deliberately chose not to use them as that would be unfair since otherwise, our proposed (efficient) pre-training method would benefit from pre-training of another language model and would prevent a fair evaluation. To make INGENIOUS self-reliant, we employed embeddings of the underlying language model as it is being pre-trained.
>
> Further, in Section 4.3, we present an empirical study on the kind of features that work the best. We found that embeddings selected from the middle layer performed better than those taken from the initial/final layers. As noted in the existing literature, initial layers capture word order; middle layers capture syntactic information, and final layers are the most specific to the pre-training objective. Semantic information, however, is spread throughout the model. Hence, based on the empirical evidence described in the paper, we hypothesize that middle-layer features capturing syntactic information are most helpful for our task. We also experimented with TF-IDF representations, although the best performance was from dense embeddings. (All of the above discussion is also presented in the first paragraph of Section 3.4, and also in Section 4.3).
>
> **Comparison to Computer Vision settings:**
>
> >Including a comparison to one of the methods mentioned in the computer vision setting would have been more useful than comparing to, e.g. loss-based sampling. I understand that these are not always applicable and typically require a supervised set-up, but some of them can probably be adapted to language tasks relatively easily.
>
> There is a fundamental difference between how subset selection techniques are employed in Computer Vision (CV) -- these techniques are used for training for specific downstream tasks (classification, tagging, etc.) and not for pre-training of the model (as we propose in this work). Given the scale of pre-training data in typical LMs, adapting techniques used for CV is practically infeasible, and addressing this challenge is, in fact, one major contribution of our work -- how to make these techniques work for LMs. To explain it further, there are two primary inefficiencies with the current subset selection methods in CV, especially when it comes to pre-training large language models:
>
> 1. **Gradient-Based Computations**: Traditional CV subset selection often involves gradient calculations, generally by retaining the model weights while only adjusting the final MLP layer used for classification. This gradient-based approach is computationally demanding as it requires calculating the per-sample loss gradients for the entire dataset. Such a process is more resource-intensive than the simple forward pass our framework uses to compute sample embeddings (Section 3.4).
>
> 2. **Need to Select Multiple Subsets for Pre-Training**: For optimal data exploration (for knowledge retention) and to reduce the risk of overfitting (for generalization), it is essential to pre-train large language models on various data subsets, which are updated regularly. With existing CV methods, this demands that the whole subset selection, including the gradient computation, be carried out multiple times during the pre-training phase. This repetitive computation further compromises the efficiency. In contrast, after establishing the initial probability distribution, the INGENIOUS framework can swiftly sample new subsets in virtually no time (Section 3.4).
>
> Furthermore, to our understanding, even within CV contexts, there has not been an exploration of pre-training using the transformer architecture with data subset selection methods.
>
> **Subset size ablation:**
>
> > Not sure I understand why increasing the subset size in some cases actually hurts performance? e.g., Table 4. I think the paper could benefit by elaborating on this, and why some subsets of the dataset are actually harmful to model performance.
>
> Great question! The size of the subset can impact the model in various ways. Note that we are trying to achieve a delicate balance between the information and "noise" that is fed to the model. While intuitively increasing the subset size provides more information to the model, it may also add noise to the model, possibly deteriorating the performance. In Table 4, we only report our empirical observations on the performance with varying subset size. We also note that recent studies ([1], [2]) have found the pre-training corpus quality, rather than just the quantity, to impact the model performance significantly.
>
> [1] Penedo, G., Malartic, Q., Hesslow, D., Cojocaru, R., Cappelli, A., Alobeidli, H., Pannier, B., Almazrouei, E. and Launay, J., 2023. The RefinedWeb dataset for Falcon LLM: outperforming curated corpora with web data, and web data only. arXiv preprint arXiv:2306.01116.
>
> [2] Touvron, H., Martin, L., Stone, K., Albert, P., Almahairi, A., Babaei, Y., Bashlykov, N., Batra, S., Bhargava, P., Bhosale, S. and Bikel, D., 2023. Llama 2: Open foundation and fine-tuned chat models. arXiv preprint arXiv:2307.09288.

---

### Meta-Review · Area_Chair_Wijm · 2023-09-19

**Recommendation:** 3

**Metareview:**

The paper under review offers a novel method, INGENIOUS, focused on the efficient selection of a subset of data for pretraining language models. This approach ensures performance comparable to training on the entire dataset. Leveraging a submodular maximization problem with a similarity-based facility location, the technique identifies representative samples from a broader corpus. When assessed, models pretrained on these subsets showcased remarkable efficiency, often matching or even surpassing models trained on full datasets in downstream tasks. The research stands out due to its rigorous experimental design, innovation in methodology, and addressing a critical issue of training efficiency. While there were concerns related to the clarity of specific results, the main focus on dataset size over other modalities, and the breadth of model validation, it is important to note that some of these criticisms may not directly align with the paper's primary focus.

Pros:
1. The INGENIOUS approach represents a significant advancement in efficient language model pretraining by extracting crucial information subsets.
2. Extensive experimental design appreciated by multiple reviewers, encompassing various architectures, sampling methods, and evaluations beyond just GLUE scores.
3. The research offers profound insights into a pivotal area, potentially paving the way for future investigations.

Cons:
1. Some ambiguities in results, especially related to the advantages of subsetting over full datasets.
2. Results on smaller models may not generalize to larger models

---

### Decision · Program_Chairs · 2023-10-07

**Decision:**

Accept-Findings

**Comment:**

The paper under review offers a novel method, INGENIOUS, focused on the efficient selection of a subset of data for pretraining language models. This approach ensures performance comparable to training on the entire dataset. Leveraging a submodular maximization problem with a similarity-based facility location, the technique identifies representative samples from a broader corpus. When assessed, models pretrained on these subsets showcased remarkable efficiency, often matching or even surpassing models trained on full datasets in downstream tasks. The research stands out due to its rigorous experimental design, innovation in methodology, and addressing a critical issue of training efficiency. While there were concerns related to the clarity of specific results, the main focus on dataset size over other modalities, and the breadth of model validation, it is important to note that some of these criticisms may not directly align with the paper's primary focus.

Pros:
1. The INGENIOUS approach represents a significant advancement in efficient language model pretraining by extracting crucial information subsets.
2. Extensive experimental design appreciated by multiple reviewers, encompassing various architectures, sampling methods, and evaluations beyond just GLUE scores.
3. The research offers profound insights into a pivotal area, potentially paving the way for future investigations.

Cons:
1. Some ambiguities in results, especially related to the advantages of subsetting over full datasets.
2. Results on smaller models may not generalize to larger models